# Parameters of Glucose Homeostasis in the Recognition of the Metabolic Syndrome in Young Adults with Prader–Willi Syndrome

**DOI:** 10.3390/jcm10235635

**Published:** 2021-11-29

**Authors:** Graziano Grugni, Antonio Fanolla, Fiorenzo Lupi, Silvia Longhi, Antonella Saezza, Alessandro Sartorio, Giorgio Radetti

**Affiliations:** 1Experimental Laboratory for Auxo-Endocrinological Research & Division of Auxology, Istituto Auxologico Italiano, IRCCS, 28824 Verbania, Italy; a.saezza@auxologico.it (A.S.); sartorio@auxologico.it (A.S.); 2Observatory for Health Provincial Government South Tyrol, 39100 Bolzano, Italy; Antonio.Fanolla@provinz.bz.it; 3Department of Pediatrics, Regional Hospital of Bolzano, 39100 Bolzano, Italy; fiorenzolupi@gmail.com (F.L.); silvia.longhi@sabes.it (S.L.); 4Marienklinik, 39100 Bolzano, Italy; giorgio.radetti@gmail.com

**Keywords:** obesity, Prader–Willi syndrome, 1 h post-load glucose, 2 h post-load glucose, metabolic syndrome, HOMA-IR, ODI, IGI

## Abstract

To verify the accuracy of different indices of glucose homeostasis in recognizing the metabolic syndrome in a group of adult patients with Prader–Willi syndrome (PWS), 102 PWS patients (53 females/49 males), age ±SD 26.9 ± 7.6 yrs, Body Mass Index (BMI) 35.7 ± 10.7, were studied. The following indices were assessed in each subject during an oral glucose tolerance test (OGTT): 1 h (>155 mg/dL) and 2 h (140–199 mg/dL) glucose levels, the oral disposition index (ODI), the insulinogenic index (IGI), the insulin resistance (HOMA-IR) were evaluated at baseline, 1 h and 2 h. Although minor differences among indices were found, according to the ROC analysis, no index performed better in recognizing MetS. Furthermore, the diagnostic threshold levels changed over the years and therefore the age-related thresholds were calculated. The easily calculated HOMA-IR at baseline may be used to accurately diagnose MetS, thus avoiding more complicated procedures.

## 1. Introduction

The metabolic syndrome (MetS) often complicates overweight/obesity, leading to atherosclerotic cardiovascular disease and type 2 diabetes mellitus (T2DM). MetS is a cluster of at least three of five medical conditions: central obesity, arterial hypertension, altered glucose metabolism, hypertriglyceridemia, and low HDL-C levels [1]. From a pathophysiological point of view, obesity and insulin resistance (IR) are believed to play a central role in the development of MetS [1,2]. Early recognition of MetS allows an intensive treatment of these patients, avoiding the worst consequences. In this context, several indices have been developed and tested in order to predict the presence of MetS both in children and adults, mainly those related to body composition and adiposity [3,4], as well as those derived from the oral glucose tolerance test (OGTT). Among the latter, post-load glucose levels after two hours (2 h PG) >140 mg/dL and <200 mg/dL during OGTT appear to be very sensitive in defining the presence of MetS [5]. More recently, it has been demonstrated that post-load glucose after one hour (1 h PG) >155 mg/dL during OGTT seemed to be more accurate in detecting MetS than the 2 h PG [6].

As in the general population, the main role that weight excess plays on individual metabolic risk clustering has also been confirmed in subjects suffering from Prader–Willi syndrome (PWS), the most common form of syndromic obesity [7,8]. PWS is a rare genetic disorder due to the lack of expression of the paternally-derived chromosomal region 15q11-13 [9]. These patients are characterized by neonatal failure to thrive, followed by worsening hyperphagia with progressive development of severe obesity, unless eating is not promptly restricted [10]. PWS patients seem to benefit from a healthier metabolic profile than essential obesity, due to their preferential subcutaneous fat distribution [11]. In spite of this, PWS subjects show a high prevalence of altered glucose metabolism, including T2DM, which appears to be more common in obese and adult subjects [12]. The exact mechanism of glucose metabolism alterations in PWS still remains unclear. Higher insulin sensitivity (with lower fasting insulin and HOMA-IR) has been reported in PWS patients in comparison with non-syndromic obese controls [13]. Other authors, however, failed to observe significant differences in insulin levels and IR between obese PWS and BMI-matched controls at all ages [14]. Furthermore, a clear relationship between obesity and insulin levels is still detectable in PWS children, with obese individuals showing higher insulin levels and HOMA-IR than non-obese subjects [7]. These discrepancies might be related to the different clinical characteristics of the study groups, including age, degree of obesity, body composition and different number of patients treated with recombinant GH and/or sex steroids.

Overall, the frequency of MetS in children and adults with PWS has been reported to be approximately 7% and 34%, respectively [7,8]. It is of note, however, that MetS is clearly linked to obesity in PWS patients. According with this observation, a similar prevalence of MetS compared to obese controls has been previously reported both in obese pediatric and adult subjects with PWS [7,8], this finding being probably one of the risk factors responsible for their excessive morbidity and mortality [15,16].

With this background, the objective of the present study was to verify the accuracy of 1 h PG vs. 2 h PG in identifying MetS in a large cohort of adult patients with PWS. In addition, we tested the accuracy of detecting the MetS of a group of other metabolic indices, such as the oral disposition index (ODI) and the insulinogenic index (IGI), which both reflect insulin secretion, and insulin resistance was evaluated with the homeostasis model assessment (HOMA-IR) at baseline, after 1 h, and after 2 h of the OGTT.

## 2. Materials and Methods

### 2.1. Study Population

A retrospective cohort study based on 102 subjects with PWS (53 females, 49 males, aged 26.9 ± 7.6 years (mean ± SD) (range 18.0–50.1), and BMI 35.7 ± 10.7 (17.2–70.9)), consecutively enrolled in the obesity inpatient clinic of the Istituto Auxologico Italiano, Piancavallo, Verbania, Italy between January 2017 and January 2020 was performed. All but one female were Caucasian. Seventy-eight subjects had interstitial deletion of the proximal long arm of chromosome 15 (del15q11-q13) (DEL15), while maternal uniparental disomy for chromosome 15 (UPD15) was found in the remaining twenty-four patients.

At the time of the study, 11 subjects (6 females) were treated for arterial hypertension. One female was undergoing therapy for hyperlipidemia, while no patients took antidiabetic drugs. Six subjects suffered from central hypothyroidism (3 females) and were biochemically euthyroid on thyroxine substitution. Twenty-eight females and nine males took sex steroid replacement therapy. Behavioral abnormalities were present in all subjects, and 50 of them (20 females) were treated with neuroleptics. Twenty-two patients were treated with recombinant GH (8 females), and fifty-eight had been treated in pediatric age. Forty-four had never received GH therapy.

The study protocol was approved by the Ethical Committee of the Istituto Auxologico Italiano (ref. no. 01C025; acronym: PWSIPMET). At the admission to our Institute, a written informed consent for the use of all biochemical and anthropometric parameters collected during hospitalization was obtained from the parents or legal guardians, and from the patients when applicable. The study was performed in accordance with the Declaration of Helsinki and with the 2005 Additional Protocol to the European Convention of Human Rights and Medicine concerning Biomedical Research.

### 2.2. Anthropometric Data

Physical examination included determination of height, weight, and waist circumference (WC) by the same trained operators, according to the Anthropometric Standardization Reference Manual [17]. Body Mass Index (BMI) was defined as weight in kilograms divided by the square of height in meters. Standing height was determined by a Harpenden Stadiometer (Holtain Limited, Crymych, Dyfed, UK). Body weight was measured to the nearest 0.1 kg, using an electronic scale (Ro WU 150, Wunder Sa.bi., Trezzo sull’Adda, Italy). WC was determined in standing position midway between the lowest rib and the top of the iliac crest after gentle expiration, with a non-elastic flexible tape measure.

### 2.3. Blood Pressure Measurements and Instrumental Examinations

Diastolic and systolic blood pressure (BP) were measured to the nearest 2 mmHg in the supine position after 5 min rest, using a standard mercury sphygmomanometer with appropriately sized cuff. The average of three measurements on different days was used.

### 2.4. Laboratory Analyses

The subjects were evaluated after a 12 h overnight fast. Baseline blood samples were drawn by venipuncture for determination of glycaemia, insulin, glycated hemoglobin (HbA1c), high-density lipoprotein cholesterol (HDL-C), and triglycerides (TG). Routine laboratory parameters were measured by enzymatic methods (Roche Diagnostics, Mannheim, Germany). In all patients, a standard OGTT (1.75 g of glucose/kg body weight up to 75 g with blood samples taken at 0, 30, 60, 90, 120 min) was performed to evaluate glucose homeostasis.

The following parameters were calculated:-altered 1 h PG = glycemia >155 mg/dL (>8.6 mmol/L);-impaired glucose tolerance (IGT) = 2 h PG between 140 to 199 mg/dL (7.8 mmol/L to 11.0 mmol/L);-insulin resistance, evaluated by the use of the homeostasis model assessment: HOMA-IR = (insulin (mIU/L) × glucose (mg/dL))/405 (Matthews), at time 0, 60 and 120 min during OGTT;-insulin sensitivity, evaluated by the use of the homeostasis model assessment: HOMA-S = 1/HOMA-IR [18];-insulin secretion, assessed with the insulinogenic index (IGI), which is the ratio of the changes in insulin (I) and glucose (G) concentration from 0 to 30 min (ΔI0–30/ΔG0–30) [19];

ß-cell compensatory capacity to change insulin sensitivity (ß-CIS), evaluated by the oral disposition index (ODI), defined as the product of HOMA-S and IGI [20].

### 2.5. Definitions

Patients were considered obese, overweight, and normal-weight with BMI values > 30, between 25–30, and <25, respectively [21].

According to the literature [22], patients were considered suffering with MetS in presence of three abnormal findings out of the following five parameters: central obesity, high systolic BP and/or diastolic BP, high triglycerides, low HDL-C, and altered glucose metabolism (AGM). Central obesity was defined when WC was >94 cm for men and >80 cm for women [23]. Hypertension was defined in presence of systolic BP values >130 mmHg and/or diastolic BP values >85 mmHg, or in case of antihypertensive drug use. Hypertriglyceridemia was defined in presence of triglycerides values >150 mg/dL or in case of a specific treatment. Low HDL-C level was defined with values <40 mg/dL in males and <50 mg/dL in females. A diagnosis of AGM was defined according to the American Diabetes Association criteria (IFG: fasting plasma glucose (FPG) 100 mg/dL (5.6 mmol/L) to 125 mg/dL (6.9 mmol/L); IGT: 2 h PG (plasma glucose) in the 75-g OGTT 140 mg/dL (7.8 mmol/L) to 199 mg/dL (11.0 mmol/L); HbA1c 5.7–6.4% (39–47 mmol/mol); T2DM: FPG levels > 126 mg/dL (>7.0 mmol/L) or 2 h PG > 200 mg/dL (>11.1 mmol/L) during an OGTT or HbA1c > 6.5% (48 mmol/mol)), or use of antidiabetic drugs [24].

### 2.6. Statistical Analysis

Statistical power analysis was performed using the PASS 14 Power Analysis software and Sample Size Software (2015). NCSS, LLC., Kaysville, UT, USA, ncss.com/software/pass, accessed on 18, February, 2020). It has been calculated that a sample of 34 patients with MetS allows a one-tailed non-inferiority test to be used for the difference between correlated proportions with a power of 80%, assuming: (i) a significance level of 5%, (ii) a proportion of the test of reference equal to 0.80, (iii) a maximum difference allowed between proportions (range of non-inferiority) equal to 0.1, (iv) a real difference between proportions equal to 0.06. Since the expected prevalence of MetS among PWS patients is 34% [8], a sample of 100 patients should identify 34 cases of MetS and 66 without MetS.

The data were first scrutinized for outliers, using a cutoff of 4.5 standard deviations. No data were excluded on this basis. To explore the data, preliminary analyses were performed. Continuous data are presented as the mean (SD) or with 95% CIs. Mean values were tested for statistical significance using 2-tailed *t* tests. Categorical data are presented as % (SE) or with 95% CIs and evaluated with Fisher’s exact test. Patients were initially classified into two groups based on altered 1 h PG and 2 h PG values. Descriptive statistics and comparison statistics were generated for the different indices and clinical conditions based on these groups.

To calculate the growth pattern of the glucose homeostasis indexes, a quantile regression was used [25] as alternative of LMS Method [26]. The logarithm of each index was used as response, fitted with a parametric model which involved age. The standardized residuals were retained to represent age-adjusted values.

Receiver operating characteristic (ROC) curves were then generated to obtain the values of area under the curve (AUC) with 95% CI, sensitivity, and specificity, for each age-adjusted standardized glucose homeostasis index as predictor of MetS [27].

In order to identify the optimal cutoff, the Youden index25 was calculated. The corresponding percentile value for each cutoff was used in the quantile regression to identify the age-specific glucose homeostasis cutoff.

Sensibility, specificity, PPV and NPV of 1 h PG and 2 h PG as predictor of MetS were calculated with both the normally used alteration levels (155 mg/dL and 140 mg/dL) and the cutoff values resulting from our analysis.

The significance threshold was set at *p* < 0.05. Bonferroni test was used for multiple comparison. The data were analyzed using SAS Enterprise Guide 4.3 (SAS Institute Inc., Cary, NC, USA).

## 3. Results

According to BMI cutoffs, 17 PWS subjects were of normal weight (16.7%, 8 females), 19 were overweight (18.6%, 8 females), and 66 were obese (64.7%, 37 females). All obese subjects had central obesity, while WC was >80 in 10 non-obese females (4 of normal weight and 6 overweight) and >94 in 9 non-obese males (1 of normal weight and 8 overweight). Thirty-two patients (16 females) had arterial hypertension. Furthermore, 10 PWS patients had hypertriglyceridemia (3 females), while 38 individuals had low HDL-C level (23 females). Additionally, 3 subjects had T2DM (2 females), 28 had impaired glucose tolerance (IGT) (16 females), and 2 had impaired fasting glucose (IFG) (1 female). The presence of MetS was detected in 29 subjects (15 females) (28.4%), without differences between the sexes. Non-obese PWS subjects had a lower frequency of MetS (6/36: 16.6%) as compared with the obese group (23/66: 34.8%), however this did not reach statistical significance (*p* = 0.1). Previous and current GH therapy did not influence the presence of MetS in the study group (data not shown).

The clinical findings and the laboratory characteristics of the entire study group, subdivided according to the absence/presence of MetS, are reported in Table 1. As expected, all parameters were significantly worse in patients with MetS than in those without MetS, with the exception of HOMA-IR value at 60 min post-OGTT, IGI and ODI.

The clinical and biochemical characteristic of PWS patients are reported in Table 2, subdivided according to normal or altered glycemic response at 1 h and 2 h. Twenty-seven PWS (26.5%) had a 1 h PG > 155 mg/dL, while 30 PWS (29.4%) showed 2 h PG > 140 mg/dL. An altered glycemic status at 2 h identifies a worse metabolic profile, as reflected by a higher HBA1c, higher triglyceride levels, and higher HOMA-IR at time 0′, 60′ and 120′. On the other hand, all parameters of MetS were similar in patients with normal 1 h PG to those with altered glycemic status.

In Table 3, sensitivity, specificity, and positive (PPV+) and negative predictive values (NPV-) are reported. As far as sensitivity is concerned, only HOMA-IR 1 h (72.4%) showed a higher sensitivity compared to ODI (53.6%) *p* < 0.05), while higher specificity was observed for ODI (70.8%) vs. HOMA-IR 1 h (50.7%), for HOMA-IR 2 h (78.1%) vs. HOMA-IR 1 h (50.7%), and for HOMA-IR 2 h (78.1%) vs. IGI (61.1%) (*p* < 0.05 for all). No statistical difference was observed for the positive and negative predictive values.

In Table 4, the area under the ROC curve of the different indices and the relative cut-off are reported. Although minor differences were observed, no index performed better. It is of note that ROC analysis showed a threshold glycemic level of 131 mg/dL at 1 h as the optimal cut-off for identifying MetS.

Finally, the age-related threshold for all indices that identify MetS are reported in Figure 1, showing their decrease over the years, with the exception of 2 h PG.

## 4. Discussion

Complications associated with morbid weight excess are recognized as the main risk factors for death throughout the life span of subjects with PWS [28]. The causes of premature mortality, particularly in adult patients, are mainly related to cardiovascular and respiratory problems, as well as to comorbidities associated with T2DM [29]. In this context, a high prevalence of MetS has been previously reported at all ages [7,8]. Thus, it is conceivable that MetS may be involved in the development of life-threatening diseases in PWS patients. Early identification and therapeutic intervention of obesity-related diseases, including MetS, have a favorable impact on the course of the syndrome, representing key factors in the management of these patients [28,29].

In this study, our aim was to find out an accurate biochemical index which might allow early identification of the risk of MetS in adult patients with PWS.

Because of the main role played by insulin in the pathogenesis of MetS [2,30], different indices of glucose homeostasis were considered as a potential mirror of the MetS.

For this purpose, we have calculated for some of the most used indexes a threshold value for age, comparing thereafter all indexes with each other, in order to check their accuracy, sensitivity, and specificity predictive values in identifying the MetS.

Our results show that sensitivity was similar for all the indices, with only HOMA-IR 60′ performing better than ODI, while HOMA-IR at 2 h showed the best specificity. Comparing the different indices with each other, no significant differences in identifying MetS were detected. As a matter of fact, the simple evaluation of basal insulin resistance, i.e., HOMA-IR at baseline, showed the same performance as the 2 h PG post-OGTT. This is a relevant new finding, very helpful for the simple detection (via single venipuncture) of MetS in these patients, allowing a two hour test to be avoided, as well as all the stress-related problems of multiple samplings. In this context, PWS patients are often disturbed by a protracted investigation, such as the OGTT, and will therefore welcome this novel approach.

We also assessed whether the thresholds of the different indices for discovering MetS presence changed over the years, observing a general decline with the progression of age, apart from 2 h PG which remained relatively stable. Therefore, the appropriate age-related cut-off for each index should be used in order to identify the patients at risk more precisely. As an additional finding, we observed that in PWS subjects, a lower glycemic level (131 mg/dL) at 1 h than the proposed one (155 mg/dL) was able to detect MetS. Perhaps this observation might be underpinned by a different metabolic milieu in these subjects, which are affected by MetS at a less deranged glycemic status.

A strength of this study is that all patients involved in the study were recruited and followed by a single center, with the same well-trained medical staff and the same laboratory, which makes the interpretation of the data more reliable than those obtained with a multicenter study. On the other hand, we acknowledge, as a weakness, that the number of the subjects taking part of the study is relatively low, thus resulting in limited strength of the statistical analysis. However, it must be considered that PWS is a rare disease, and enrolment of these patients is extremely difficult. Another limitation is the cross-sectional design of the study, which does not allow us to draw conclusions about the natural history of MetS in PWS, particularly on the time of its appearance and its development during the life span of these subjects. In this context, more research will be necessary to assess the development of MetS during adulthood in PWS, more specifically in older ages. A further weakness of our study is that it was conducted in Caucasian patients, and its findings may not be extended to other ethnic groups.

Notwithstanding these limitations, our results suggest that a simple index such as HOMA-IR can contribute to detecting the presence of MetS in adults with PWS, as well as other indices that require more complicated diagnostic procedures. In a previous work by our group, we recently demonstrated that BMI, the simplest and most commonly used anthropometric index, performs as good as other indexes, which also consider body composition and fat distribution, in recognizing the presence of MetS in PWS [31]. Overall, from a practical point of view, our findings suggest that the indices that are the simplest to calculate should be preferred to detect MetS in these patients, because they allow procedures to be minimized that the patient has to go through, facilitating the work of clinicians, speeding up diagnostics, and leading to cost reductions.

In conclusion, our study showed that the simple basal evaluation of insulin resistance was significantly associated with the presence of MetS in adult subjects with PWS, thus avoiding more complicated procedures, including the OGTT, promoting patient compliance. The results of the present study could be useful for clinicians, optimizing the diagnostic and therapeutic management of PWS, as well as for increasing awareness of the syndrome.

## Figures and Tables

**Figure 1 jcm-10-05635-f001:**
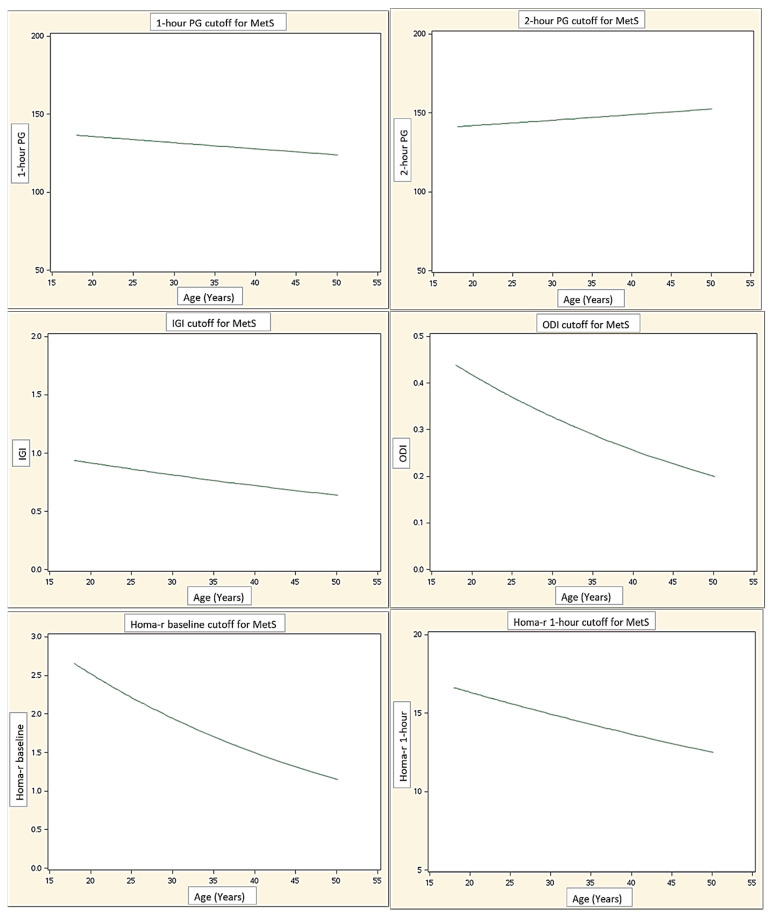
Age-related threshold for all indices, beyond which the risk of MetS is identified. Abbreviations: PG: post-load glucose at 1 h and 2 h post-OGTT; IGI: insulinogenic index; ODI: oral disposition index; HOMA-IR: insulin resistance at baseline, 1 h and 2 h post-OGTT.

**Table 1 jcm-10-05635-t001:** Clinical and laboratory characteristics of the study group. For significance: ^#^ *p* < 0.05; * *p* <0.01; ^°^ *p* < 0.001; ^§^ *p* < 0.0001.

	Total	MetS−	MetS+
Clinical data			
Number of subjects	102	73	29
Age (years)	26.9 ± 7.6	26.5 ± 7.5	28.1 ± 7.9
Female sex, n (%)	53 (52%)	38 (52%)	15 (51.7%)
BMI (kg/m^2^)	35.7 ± 10.7	33.6 ± 9.8	41.1 ± 11.2 *
WC (cm)	106 ± 18.3	102.2 ± 16.8	115.6 ± 18.9 ^°^
WC (cm) females	103.3 ± 19.6 (no. 53)	99.1 ± 17.1 (no. 38)	113.9 ± 21.9 ^#^ (no. 15)
WC (cm) males	108.9 ± 16.6 (no. 49)	105.5 ± 15.9 (no. 35)	117.4 ± 15.7 ^#^ (no. 14)
SBP (mm/Hg)	119.6 ± 8.8	117.0 ± 8.0	126.2 ± 7.3 ^§^
DBP (mm/Hg)	75.8 ± 6.5	74.7 ± 6.4	78.6 ± 6.0 *
Laboratory data			
HDL-C (mg/dL)	49.7 ± 12.4	52.8 ± 12.3	42.0 ± 9.2 ^§^
HDL-C (mg/dL) females	53.2 ± 12.5 (no. 53)	56.1 ± 12.6 (no. 38)	45.8 ± 9.1 * (no. 15)
HDL-C (mg/dL) males	46.0 ± 11.3 (no. 49)	49.2 ± 11.0 (no. 35)	37.9 ± 7.7 * (no. 14)
TG (mg/dL)	95.1 ± 43.1	87.1 ± 32.6	115.2 ± 58.2 *
glycemia (mg/dL) T0	84.6 ± 9.6	82.4 ± 7.7	90.3 ± 11.5 ^§^
insulin (mIU/L)	9.9 ± 7.3	8.5 ± 5.8	13.5 ± 9.2 ^°^
HbA1c (%)	5.4 ± 0.4	5.3 ± 0.3	5.7 ± 0.5 ^§^
Metabolic indices			
glycemia (mg/dL) T60	137.2 ± 33.7	131.8 ± 30.2	150.9 ± 38.5 *
glycemia (mg/dL) T120	125.0 ± 32.6	118.3 ± 27.4	141.7 ± 38.7 ^°^
HOMA-IR T0	2.15 ± 1.70	1.79 ± 1.32	3.07 ± 2.16 ^°^
HOMA-IR T60	22.86 ± 25.74	20.27 ± 16.59	29.40 ± 40.25
HOMA-IR T120	22.22 ± 20.87	18.95 ± 18.25	30.45 ± 24.82 ^#^
IGI	0.86 ± 1.62	0.76 ± 1.85	1.09 ± 0.78
ODI	0.53 ± 1.99	0.48 ± 2.26	0.66 ± 1.03

Abbreviations: BMI: Body Mass Index; WC: waist circumference; SBP: systolic blood pressure; DBP: diastolic blood pressure; HDL-C: high-density lipoprotein cholesterol; TG triglycerides; HbA1c: glycated hemoglobin; HOMA-IR: insulin resistance; T0: time 0 min in the OGTT; T60: time 60 min after the OGTT; T120: time 120 min after the OGTT; IGI: insulinogenic index; ODI: oral disposition index.

**Table 2 jcm-10-05635-t002:** Clinical and biochemical data in the PWS patients, subdivided according to the glucose value at 1 h and 2 h of OGTT. For significance: ^#^ *p* < 0.05; * *p* < 0.01; ^§^ *p* < 0.0001.

	Glycemic Response at 1 h		Glycemic Response at 2 h	
	Normal	Altered	Normal	Altered
Number of subjects	75	27	72	30
Age (years)	27.2 ± 7.8	26.2 ± 7.3	26.8 ± 7.2	27.2 ± 8.6
Female/male	37/38	16/11	36/36	17/13
BMI (kg/m^2^)	35.1 ± 10.7	37.4 ± 10.6	35.4 ± 10.8	36.4 ± 10.7
WC (cm)	105.0 ± 18.1	108.7 ± 19	105.7 ± 18.8	106.6 ± 17.6
SBP (mmHg)	118.8 ± 9.1	121.9 ± 7.9	118.6 ± 8.9	122.0 ± 8.5
DPB (mmHg)	75.5 ± 6.8	76.5 ± 5.5	75.6 ± 6.4	76.2 ± 6.7
HDL-C (mg/dL)	49.3 ± 12.5	50.8 ± 12.4	49.3 ± 12.7	50.7 ± 12
TG (mg/dL)	91.7 ± 44.2	104.7 ± 39.2	89.3 ± 41.4	108.9 ± 44.6 ^#^
1 h PG	121.2 ± 20.5	181.6 ± 20.7 ^§^	124.1 ± 26.3	168.7 ± 28.6 ^§^
2 h PG	112.8 ± 22.7	158.7 ± 32.2 ^§^	108.9 ± 20.1	163.4 ± 23 ^§^
HbA1c (%)	5.38 ± 0.34	5.59 ± 0.55	5.4 ± 0.4	5.6 ± 0.4 *
HOMA-IR T0	2.0 ± 1.5	2.7 ± 2.2	1.8 ± 1.4	2.9 ± 2 ^#^
HOMA-IR T60	16.2 ± 11.9	41.5 ± 41	17.9 ± 14.6	34.9 ± 39.7 ^#^
HOMA-IR T120	15.76 ± 14.48	40.17 ± 25.3	13.25 ± 8.04	43.74 ± 26.06 ^§^
IGI	0.89 ± 1.87	0.75 ± 0.42	0.85 ± 1.89	0.87 ± 0.61
ODI	0.57 ± 2.31	0.43 ± 0.36	0.60 ± 2.36	0.38 ± 0.26

Abbreviations: BMI: Body Mass Index; WC: waist circumference; SBP: systolic blood pressure; DBP: diastolic blood pressure; HDL-C: high-density lipoprotein cholesterol; TG triglycerides; 1 h PG: post-load glucose at time 60 min post-OGTT; 2 h PG: post-load glucose at time 120 min post-OGTT; HbA1c: glycated hemoglobin; HOMA-IR: insulin resistance; T0: time 0 min of OGTT; T60: time 60 min post-OGTT; T120: time 120 min post-OGTT; IGI: insulinogenic index; ODI: oral disposition index.

**Table 3 jcm-10-05635-t003:** Sensitivity, specificity, positive and negative predictive values for the different indices for identifying MetS. For significance: * *p* < 0.05 vs. ODI; ^$^ *p* < 0.05 vs. HOMA-IR T60; ^£^ *p* < 0.05 vs. HOMA-IR T60; ^&^ *p* < 0.05 vs. IGI.

	Sensitivity	Specificity	PPV	NPV
IGI	57.1% (47.9–66.4)	61.1% (52.0–70.3)	36.4% (27.3–45.4)	78.6% (70.9–86.3)
ODI	53.6% (44.2–62.9)	70.8% ^$^ (62.3–79.4)	41.7% (32.4–50.9)	79.7% (72.1–87.2)
HOMA-IR T0	69.0% (60.3–77.7)	68.5% (59.8–77.2)	46.5% (37.1–55.9)	84.7% (78.0–91.5)
HOMA-IR T60	72.4% * (64.0–80.8)	50.7% (41.3–60.1)	36.8% (27.8–45.9)	82.2% (75.0–89.4)
HOMA-IR T120	58.6% (49.4–67.9)	78.1% ^£,&^ (70.3–85.5)	51.5% (42.1–60.9)	82.6% (75.5–89.7)

Abbreviations: PPV: positive predictive value; NPV: negative predictive value; IGI: insulinogenic index; ODI: oral disposition index; HOMA-IR: insulin resistance; T0: time 0 min of the OGTT; T60: time 60 min post-OGTT; T120: time 120 min post-OGTT.

**Table 4 jcm-10-05635-t004:** The area under the receiver operating characteristic (ROC) curve of the different indices and the relative cut-off for identifying MetS.

	AUC	Cut-Off
1 h PG	0.65 (0.52–0.77)	131.1
2 h PG	0.70 (0.57–0.82)	141.0
HOMA-IR T0	0.69 (0.57–0.81)	2.0
HOMA-IR T60	0.60 (0.48–0.72)	14.6
HOMA-IR T120	0.68 (0.56–0.80)	26.1
IGI	0.59 (0.47–0.71)	0.8
ODI	0.60 (0.47–0.73)	0.4

Abbreviations: 1 h PG: post-load glucose at time 60 min during OGTT; 2 h PG: post-load glucose at time 120 min post-OGTT; HOMA-IR: insulin resistance; T0: time 0 min of the OGTT; T60: time 60 min post-OGTT; T120: time 120 min post-OGTT; IGI: insulinogenic index; ODI: oral disposition index.

## Data Availability

The raw data are available at https://doi.org/10.5281/zenodo.5735432.

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
