# Peer review of "Parameters of Glucose Homeostasis in the Recognition of the Metabolic Syndrome in Young Adults with Prader–Willi Syndrome"

_jcm, 2021, doi:10.3390/jcm10235635_

Round 1

Reviewer 1 Report

The authors present a highly clinically relevant topic to improve diagnostic workload in diagnostics in subjects with PWS. 

I have a few comments:

Introduction: the terms MetS and disorders of glucose homeostasis are used as if they are interchangeable, while impaired glucose tolerance conditions are on of the factors in MetS. Can you make this clearer? Readability would improve if the definition of MetS is stated in the introduction, and not only in the methods. 

Methods: statistical power calculations are lacking. 

Results: it is not clear why for some data female and male specific outcomes are reported, while this is not for other outcomes in Table 1. Did you detect differences in subjects on growth hormone and those that were not? Can you elaborate on the effect of growth hormone, even if none of the subjects is currently GH supplemented?

The layout of Table 1 could be improved to separate the laboratory characteristics better and make data presentation clearer.

Discussion:

If HOMA-IR indicates insulin resistance, what would be your next step? Can you do without an additional OGTT in all subjects? 

In the conclusion you state that MetS can be diagnosed based on evaluation of insulin resistance, while the definition of MetS is much broader and for  diagnosis at least 3 abnormal findings are needed. Can you correct this.

Author Response

Responses to reviewer #1

Manuscript ID: jcm-1473662

Thank you very much for reviewing our manuscript and for your positive comments. We wish to express our sincerest appreciation for your insightful comments on our paper. We hope that the revisions will satisfy your standard. Text changes are highlighted in bold.

Q1 Introduction: the terms MetS and disorders of glucose homeostasis are used as if they are interchangeable, while impaired glucose tolerance conditions are on of the factors in MetS. Can you make this clearer?

A1 We checked the possible interchangeable use of the two terms and found a possible overlap when referring to the presence of Mets in PWS patients. We have therefore changed the paragraph.

Q2 Readability would improve if the definition of MetS is stated in the introduction, and not only in the methods.

A2 The definition of MetS has been added in the Introduction section.

Q3 Methods: statistical power calculations are lacking

A3 Data about statistical power analysis has been added in the text (statistical analysis section).

Q4 Results: it is not clear why for some data female and male specific outcomes are reported, while this is not for other outcomes in Table 1.

A4 We differentiated HDL and WC values by sex as they have different cut-offs in males and females for the definition of MetS, unlike the other parameters taken into consideration.

Q5 Did you detect differences in subjects on growth hormone and those that were not? Can you elaborate on the effect of growth hormone, even if none of the subjects is currently GH supplemented?

A5 Previous and current GH therapy did not influence the presence of MetS in the study group. A sentence has been added in the text (Results section).

Q6 The layout of Table 1 could be improved to separate the laboratory characteristics better and make data presentation clearer.

A6 . We have clearly separated the different data (clinical data, laboratory data, metabolic indices) in the table 1. We have preferred to keep a single table instead of dividing it into three different tables.

Q7 Discussion: If HOMA-IR indicates insulin resistance, what would be your next step? Can you do without an additional OGTT in all subjects?

A7 In the last paragraph of the discussion a sentence on the use of the OGTT was added.

Q8 In the conclusion you state that MetS can be diagnosed based on evaluation of insulin resistance, while the definition of MetS is much broader and for  diagnosis at least 3 abnormal findings are needed. Can you correct this.

A8 We completely agree with your comment. The text has been changed accordingly, both in Introduction section and in Discussion section.

Reviewer 2 Report

The study aim to compare different parameters in metabolic syndrome evaluation of PWS cohort. The authors conclude that basal evaluation is enough to diagnose them. 

I have some comments:

-page 2 line 49: In pite of this...: Metabolic syndrome is clearly linked to obesity in PWS patients. This paragraph should be reviewed to clarify that.

-material: it's a retrospective study but includes consent form from the families. Please clarify

-Figure 1: charts can be seen (you should increase letter size)

-page 8 line 243: "not fully clarified".. Again MetS is related to weight and it's absent in PWS children without obesity.

-negative predictive value from HOMA basal is highest. I think you should discuss that , because in fact it's more rellevant this data that comparison with 1-2 hour data.

Author Response

Responses to reviewer #2

Manuscript ID: jcm-1473662

Thank you for your commentaries and very detailed suggestions. In the passage below, we will discuss each of your recommendations point by point in the order of occurrence. We hope that the revisions will satisfy your standard. Text changes are highlighted in bold.

Q1 page 2 line 49: In spite of this...: Metabolic syndrome is clearly linked to obesity in PWS patients. This paragraph should be reviewed to clarify that.

A1 The text has been changed accordingly.

Q2 material: it's a retrospective study but includes consent form from the families. Please clarify.

A2 Thank you for your suggestion. A sentence has been added in the text.

Q3 Figure 1: charts can be seen (you should increase letter size)

A3 Letter size in figure 1 has been increased.

Q4 page 8 line 243: "not fully clarified".. Again MetS is related to weight and it's absent in PWS children without obesity..

A4 The sentence is not clear and misleading and has been removed.

Q5 negative predictive value from HOMA basal is highest. I think you should discuss that , because in fact it's more relevant this data that comparison with 1-2 hour data.

A5 Thank you for your comment. However, no statistical difference was observed for the positive and negative predictive values of the different indices taken into consideration (please, see Table 3 and the text)
